# Identification of Time-Invariant Biomarkers for Non-Genotoxic Hepatocarcinogen Assessment

**DOI:** 10.3390/ijerph17124298

**Published:** 2020-06-16

**Authors:** Shan-Han Huang, Ying-Chi Lin, Chun-Wei Tung

**Affiliations:** 1Ph. D. Program in Toxicology, Kaohsiung Medical University, Kaohsiung 80708, Taiwan; u102832002@gmail.com (S.-H.H.); yclin@kmu.edu.tw (Y.-C.L.); 2School of Pharmacy, College of Pharmacy, Kaohsiung Medical University, Kaohsiung 80708, Taiwan; 3Research Center for Environmental Medicine, Kaohsiung Medical University, Kaohsiung 80708, Taiwan; 4Graduate Institute of Data Science, College of Management, Taipei Medical University, Taipei 11031, Taiwan; 5National Institute of Environmental Health Sciences, National Health Research Institutes, Miaoli County 35053, Taiwan

**Keywords:** time-invariant biomarkers, non-genotoxic hepatocarcinogens, toxicogenomics, machine learning

## Abstract

Non-genotoxic hepatocarcinogens (NGHCs) can only be confirmed by 2-year rodent studies. Toxicogenomics (TGx) approaches using gene expression profiles from short-term animal studies could enable early assessment of NGHCs. However, high variance in the modulation of the genes had been noted among exposure styles and datasets. Expanding from our previous strategy in identifying consensus biomarkers in multiple experiments, we aimed to identify time-invariant biomarkers for NGHCs in short-term exposure styles and validate their applicability to long-term exposure styles. In this study, nine time-invariant biomarkers, namely A2m, Akr7a3, Aqp7, Ca3, Cdc2a, Cdkn3, Cyp2c11, Ntf3, and Sds, were identified from four large-scale microarray datasets. Machine learning techniques were subsequently employed to assess the prediction performance of the biomarkers. The biomarker set along with the Random Forest models gave the highest median area under the receiver operating characteristic curve (AUC) of 0.824 and a low interquartile range (IQR) variance of 0.036 based on a leave-one-out cross-validation. The application of the models to the external validation datasets achieved high AUC values of greater than or equal to 0.857. Enrichment analysis of the biomarkers inferred the involvement of chronic inflammatory diseases such as liver cirrhosis, fibrosis, and hepatocellular carcinoma in NGHCs. The time-invariant biomarkers provided a robust alternative for NGHC prediction.

## 1. Introduction

Chemical exposure, including those from environmental, diet, and other sources, was estimated to account for about 45–50% of cancer formation [1]. The liver is the most vulnerable organ to chemicals capable of inducing cancers, and many chemicals are known to induce cancer in the liver [2,3]. Based on the pathogenic mechanism, these hepatocarcinogens can be categorized as either genotoxic or non-genotoxic hepatocarcinogens (NGHCs) [4,5]. In contrast to genotoxic hepatocarcinogens, which can be easily identified by in vitro bioassays [6], the assessment of NGHCs relies on long-term rodent bioassays [7]. Although the “gold standard” method provides the quantitative information on dose–response behavior for determining the carcinogenic potential of a chemical, it is hampered by high costs and inefficiency [7]. The development of novel well-validated short-term screening methods is therefore desirable for identifying potential NGHCs for further experimental validation.

Given the diverse mechanisms of action and organ-specificity of NGHCs, toxicogenomic (TGx) models are promising alternative approaches for assessing NGHCs and deciphering the underlying mechanism of the response [8]. Hepatic gene expression signatures derived from 5-day animal models were shown to be better biomarkers for hepatic tumor formation for NGHCs than traditional in vivo pathological and genomic biomarkers, such as liver histological changes, serum alanine aminotransferase activity, cytochrome P450 genes, and Tsc-22 or alpha2-macroglobulin messenger RNA [9]. A TGx-based model with 5-day animal data was also shown to have better predictive accuracy than quantitative structure–activity relationship (QSAR) models. To date, DrugMatrix [10], Gene Expression Omnibus accession no. 8858 (GSE8858) [11], and Toxicogenomics Project-Genomics Assisted Toxicity Evaluation System (TG-GATEs) [12] are three major large-scale datasets providing gene expression data under various NGHC exposure styles, including single or repeated low, medium, and high (maximum tolerated) doses treated for 1 day, 3 days, 5 days, 7 days, 14 days, and 28 days for TGx model development. A few TGx models were developed using biomarkers identified from single or multiple short-term NGHC microarray datasets [10,13,14,15,16,17,18,19]

Although the models performed well for NGHC assessment in their corresponding exposure styles, they derived very different biomarkers [10,13,16]. The results imply that the predictive performance of the reported biomarkers derived from one exposure style may not be useful for another exposure style. The expression of some biomarkers may vary dramatically over a short period of time, even be reversed. A biomarker whose expression varies in different timepoints may not be reliable for NGHCs prediction and mechanism interpretation. Utilizing only time-invariant biomarkers should derive a more reliable and applicable NGHC prediction model.

This study aimed to analyze the pattern of biomarkers and identify the time-invariant biomarkers for NGHCs prediction. Time-invariant biomarkers will be derived from short-term exposure styles and their prediction performance validated based on long-term exposure styles. A total of nine genes were identified as time-invariant biomarkers, including the upregulation of Akr7a3, Aqp7, Cdc2a, and Cdkn3, and downregulation of A2m, Ca3, Cyp2c11, Ntf3, and Sds. The comparison with published biomarkers showed that the time-invariant biomarkers achieved a reliable performance in various short-term exposure styles. The prediction results based on an independent test dataset further confirmed the usefulness of the time-invariant biomarkers. An enrichment analysis of the time-invariant biomarkers was conducted to provide a better inference of the underlying diseases associated with non-genotoxic hepatocarcinogenesis.

## 2. Materials and Methods

The systematic flow of the time-invariant biomarkers identification and model development analysis is shown in Figure 1

### 2.1. Chemical List and Microarray Datasets

The chemical list utilized in this study has been used in our previous work for consensus biomarkers for predicting NGHCs [16]. In brief, NGHCs and non-hepatocarcinogens (s) consistently classified by several published studies were compiled to get the largest list, and 274 chemicals (50 NGHCs and 224 NHCs) were identified.

Gene expression data from DrugMatrix, GSE8858, and TG-GATEs were analyzed. Based on the platforms utilized, DrugMatrix consists of DrugMatrix with the Affymetrix platform (DMA) and DrugMatrix with the Codelink platform (DMC). GSE8858 utilized the Codelink platform, and the TG-GATEs utilized the Affymetrix platform. The number of chemicals relevant to non-genotoxic hepatocarcinogenesis is 88 for DMA and 174 for DMC, respectively. GSE8858 is a subset of a large liver xenobiotic and pharmacological response database produced by Iconix Biosciences [20], which contains the gene expression profiles of 178 chemicals. A total of 105 chemicals from the TG-GATEs were identified as non-genotoxic hepatocarcinogens according to cytotoxic oxidative stress, one important mechanism for NGHCs. The numbers of NGHCs:NHCs in the DMA, DMC, GSE8858, and TG-GATEs were 26:62, 36:138, 39:139, and 12:93, respectively.

The experimental protocols of all four datasets were similar in animal strain (Sprague–Dawley), sex (male), age (6–8 weeks old), and environmental conditions. Each dose–exposure style experiment (in vivo bioassay) was conducted in biological triplicates. The maximum tolerated dose (MTD) was defined as a 50% reduction in weight gain over the control after a 5-day repeated dose in DrugMatrix and GSE8858. In contrast, the highest dose was set as the dose that induces the minimum toxic effect over the course of a 4-week toxicity study in TG-GATEs.

GSE8858 consists of data from a 1-day single-dose as well as 3- and 5-day repeat once-daily dose experiments at the MTD. DrugMatrix consists of data from 6 h and 1-day single dose and 3- and 5-day repeated once-daily dose experiments at the MTD, 50% of the MTD (mid), and 25% of the MTD (low). TG-GATEs consists of data from 3-, 6-, 9- and 24-h single doses, and repeated once-daily doses of the 3-, 7-, 14- and 28-day experiments at high, middle, and low doses (dose ratio 10:3:1).

To maximize the number of data for subsequent analysis from the referenced databases, we defined the MTD treatments in DrugMatrix and GSE8858 and the highest dose treatment in TG-GATEs as high-dose, and the 5-day and the 7-day exposure styles were grouped as 1-week. The exposure styles of the 1-day, 3-day, and 1-week levels were grouped as short-term exposure, while the 14-day and 28-day levels were grouped as long-term exposure.

The metadata were downloaded from the websites of DrugMatrix (ftp://anonftp.niehs.nih.gov/drugmatrix/), GSE8858 (ftp://ftp.ncbi.nlm.nih.gov/geo/series/GSE8nnn/GSE8858/) and TG-GATEs (ftp://ftp.dbcls.jp/archive/open-tggates/) and imported into the RStudio software environment (RStudio, Boston, MA, USA). All the gene expression profiles were normalized and log2-transformed for subsequent analysis.

### 2.2. Identification of the Time-Invariant Biomarker Sets

Three common exposure styles of the referenced datasets, namely the 1-day, 3-day, and 1-week high-dose exposures, were considered for the identification of the time-invariant biomarkers. First, each consensus biomarker set was identified as the overlapped differential expressed genes (DEGs) based on a t-test (*p* < 0.05), and a 1.5-fold change [16], which were derived from each common exposure style of these datasets. Subsequently, each consensus biomarker set was cross-checked with the other two exposure styles. The set of consistently up- or downregulated biomarkers in all three exposure styles was identified as the time-invariant biomarkers.

### 2.3. Model Development

Machine learning classifiers have been widely applied to model the complex relationships between biomarkers and toxicity. In this study, we employed seven well-known classifiers, including decision tree (J48) [21], bagging tree [22], boosting tree [23], k-nearest neighbor (kNN) [24], Naive Bayes (NB) [25], support vector machine (SVM) [26], and Random Forest (RF) [27], to evaluate the reliabilities of the consensus biomarkers. Published biomarker sets, including five genes from Eichner et al. 2014 (E5) [13], 19 genes from Fielden et al. 2011 (F19) [10], and nine genes from Uehara et al. 2011 (U9) [12], were utilized for comparison. E5 and U9 were obtained from the highest dose of the 14-day and 28-day exposure styles in TG-GATEs, respectively, while F19 was identified from the MTD of the 5-day exposure style in DrugMatrix.

The decision tree-based ensemble learning algorithm RF was found to perform best in our datasets. RF improves the prediction performances of decision trees and reduces variance to avoid overfitting based on a set of decision trees built on bootstrap samples from the training dataset and a fixed number of randomly selected features for tree splitting. The prediction of a given sample is based on a majority vote by the fully grown decision trees. The implementation of the RF algorithm was based on WEKA v3.8 (WEKA, Hamilton, New Zealand). The number of features for constructing a fully grown decision tree was set to the default value of the square root of the number of features and genes of each biomarker set. The optimal number of trees ranging from 10 to 100 was determined based on the AUC performance from the leave-one-out cross-validation (LOOCV). All the machine learning algorithms and LOOCV procedures were implemented using the package of WEKA. The variance interquartile range (IQR), as well as the coefficients of variances from the datasets (C.V.d) and exposures (C.V.e) were calculated to assist biomarker evaluation. IQR measured the overall variance based on the AUCs from the models with individual biomarker sets, while C.V.d and C.V.e indicated the source of the variances from the difference in the datasets and exposures, respectively. An IQR, C.V.d, or C.V.e value greater than 5% indicated that the performance of the models was not stable.

### 2.4. External Validation

The Johnson and Johnson dataset (JNJ) dataset [28], consisting of data from the single-dose 1-day experiments at the MTD for 9 NGHCs and 54 NHCs, was utilized for external validation of the developed models. For each chemical, the expression values measured based on the Codelink platform are available for analysis. The raw data were downloaded from the public database of Chemical Effects in Biological Systems [29] and were normalized and log2-transformed. Since JNJ utilized the Codelink platform, only models trained with the DMC and GSE8858 datasets, which also utilized the Codelink platform, were evaluated.

### 2.5. Enrichment Analysis

To better understand the roles of the time-invariant biomarkers, enrichment analysis of the Gene Ontology (GO), pathway, and disease terms were conducted based on the Comparative Toxicogenomics Database (CTD) [30]. In the version of August 2018, CTD includes over 2.3 million manually curated chemical–gene, chemical–phenotype, chemical–disease, gene–disease, and chemical–exposure interactions for 15,681 chemicals, 46,689 genes, 4340 phenotypes, and 7212 diseases. For further analysis and hypothesis development, CTD includes over 38 million toxicogenomic relationships, such as internal integration of these direct interactions generating over 24 million gene–disease sets that are statistically ranked and external integration with annotations from GO, Kyoto Encyclopedia of Genes and Genomes (KEGG), Reactome, and BioGRID. In the latest edition, the CTD has maintained and created MEDIC by merging the disease terms from the flat list of the Online Mendelian Inheritance in Man (OMIM) resource into the Medical Subject Headings (MESH) disease hierarchy. A corrected p-value less than 0.05 was considered as the criteria to identify the significantly enriched GO, pathway, and disease terms.

## 3. Results and Discussion

### 3.1. Time-Invariant Biomarkers and Machine Learning Classifiers

By analyzing the consensus biomarkers derived from the 1-day, 3-day, and 1-week experiments, we found that modulation of the biomarkers of the 1-day and 1-week levels were relatively less consistent than the 3-day level. The consensus biomarker set (all genes are consistently up- or downregulated) derived from the 3-day exposure style was found to be time-invariant in the short-term exposure (Table 1). In contrast, the modulation of E5, F19, and U9 varied in different exposure styles. While upregulated genes are the preferred biomarkers due to easy implementation of the diagnosis method, downregulated genes can be also useful as shown in the previous studies [31,32]

To evaluate the classification performance of the time-invariant biomarkers, we implemented seven machine learning algorithms and compared their LOOCV performance for choosing the best classifier. The RF-based models achieved the highest median AUC of 0.817 and the second-lowest variance IQR of 0.041 (Table 2). The Naive Bayes (NB) models yielded a median AUC of 0.800 with a variance IQR of 0.055. Although bagging tree (BaT) yielded the lowest variance IQR of 0.035 and its median AUC was 0.809, its C.V.d and C.V.e were both higher than 5%. Therefore, RF-based models were chosen for the following analysis. 

To better understand whether or not the time-invariant biomarkers obtained from the short-term exposure datasets are robust, we further evaluated the prediction performance of the biomarkers on all exposure styles equal or longer than 1 day. The performance is shown in Table 3. The time-invariant biomarker set (consensus 3-day biomarkers) achieved the highest median AUC of 0.824 and a low IQR of 0.036 (Table 3). Consensus 1-week biomarkers achieved a median AUC of 0.810 for all exposure styles; however, its IQR (0.111), C.V.d (7.41%), and C.V.e (7.25%) were all higher than 5%. F19 achieved a median AUC of 0.809 for all exposure styles; however, its IQR (0.085) and C.V.d (6.47%) were both much higher than the time-invariant biomarker set. Please note that the time-invariant biomarker set further improved the median AUC value by 9% compared to our previously published consensus biomarkers obtained from the 1-day exposure style (median AUC of 0.733) [16]. The results indicated that the time-invariant biomarkers can also be applied to the long-term exposure style and still provide good prediction.

### 3.2. External Validation

An independent dataset, the JNJ dataset, was applied for external validation of the time-invariant biomarkers. Table 4 presents the prediction performance of the different biomarker sets on the JNJ dataset. The models were trained with 1-day datasets of DMC or GSE8858, which are also based on the Codelink platform. For all the other published biomarkers, only the genes that can be identified in the Codelink platform were utilized. The time-invariant biomarkers achieved good performances with the highest AUC values of 0.862 and 0.857 for models based on DMC and GSE8858, respectively. The performance of the time-invariant biomarkers was better than other compared biomarker sets, even though the JNJ dataset was from 1-day experiments. Therefore, the time-invariant biomarkers are expected to be useful for identifying NGHCs regardless of exposure styles.

### 3.3. Analysis of the Time-Invariant Biomarkers

The identified nine time-invariant biomarkers are A2m, Akr7a3, Aqp7, Ca3, Cdc2a, Cdkn3, Cyp2c11, Ntf3, and Sds. A2m encodes a protease inhibitor and cytokine transporter that can inhibit a broad spectrum of proteases and inflammatory cytokines. Ca3 encodes a member of carbonic anhydrase. A2m and Ca3 had been identified as consensus NGHC biomarkers in our previous study [16] and their reduction have been associated with hepatocarcinogenicity of NGHCs [33,34].

Akr7a3 encodes an aldo–keto reductase that plays roles in the detoxification of aldehydes and ketones. It has been identified as an NGHC biomarker by published studies [10,13,17,35,36,37], in which it was upregulated by oxidative stress, a known tumor-promoting mechanism for NGHCs. The reductase level was also observed to be upregulated in rat hepatoma [38]. Aqp7 encodes a member of the aquaporin channel family, which facilitates the transport of glycerol from adipocytes to the liver. The encoded protein also allows the movement of water and urea across cell membranes. Aqp7 is a significant modulator of whole-body energy metabolism in a wide range of tissues, including in adipocytes and liver cells, in rats and humans [39]. The gene had been reported to be significantly elevated in malignant and borderline liver tumors compared to in benign tumors differentiated using rat liver slices [40], but the role of Aqp7 upregulation in liver tumor formation is still unknown.

Cyp2c11 encodes cytochrome P450 2C11 in rats, which is a functional counterpart of human Cyp2c9. The most abundant male-specific isoform of CYP in rats mediates the hydroxylation of some endogenous steroids, such as testosterone. Ntf3 encodes a neurotrophin protein that is closely associated with both nerve growth factor and brain-derived neurotrophic factor. Downregulation of Cyp2c11 and Ntf3 has been reported to play crucial roles in inflammation [41] and the AhR signaling pathway [42], respectively. These two modulations have been also observed activating following acute and subchronic exposures to NGHCs in previous studies [42]. Sds encodes the L-serine dehydratase, which is involved in the pathway gluconeogenesis; it is also a stress-associated gene that is downregulated after 24 h of treatment of hepatocarcinogens in vivo [43] and remains downregulated during the development of rat liver cancer [44].

Cdc2a, also known as Cdk1 (cyclin-dependent kinase 1), encodes a cell division control protein [45]. During liver regeneration, the essential cell cycle indicates the gene is sufficient to drive the proliferation of all cell types up to mid-gestation [45]. Cdc2a protein was reported frequently augmented in hepatocarcinoma (HCC) tissue, and such dysfunctional cell cycle regulation, which contributes to the generation of cancer stem cells, may promote tumorigenesis [46]. The signature is also one of the up-regulated DEGs promoting cirrhosis to HCC in published bioinformatics analysis [47] and is associated with the oxidative stress by exposure to diethylnitrosamine [15]. Diethylnitrosamine (DEN) is an environmental carcinogen as an initiator for hepatocarcinogenesis. After DEN short-term administration, lipid peroxidation can be detected, as well as overexpression of glutathione-S-transferase Pi (GSE-p); this is considered a marker of initiation in chemical-induced hepatocarcinogenesis. Cdkn3 encodes cyclin-dependent kinase inhibitor 3, which is involved in regulating the cell cycle. The protein acts as a cyclin-dependent kinase inhibitor that selectively binds to Cdk2 kinase to inhibit G1/S transition, as well as form a complex with Mdm2 and p53 to facilitate cell cycle progression [48]. Overexpression of Cdkn3 in HCC was correlated with poor tumor differentiation and advanced tumor stage. Cdkn3 had been reported as part of a vascular invasion signature [49]. In a previous study utilizing bioinformatics-based screening, the upregulation of Cdkn3 was also identified as a marker of transformation from cirrhosis into HCC and correlated with the occurrence, invasion, and recurrence of HCC [50]. Akr7a3 [51], Cdc2a [50,52], and Cdkn3 [50] have also been found as biomarkers for early diagnosis, staging, and prognosis in human liver cancer clinically.

To provide insights into the underlying mechanism, enrichment analysis was conducted to infer the GO, pathway, and disease terms associated with the identified biomarkers. Results show that seven disease terms were identified as significantly associated diseases with adjusted p-values < 0.05. For the GO and pathway terms, the analysis identified no significant GO or pathway terms. Table 5 lists only the inferred significant disease terms of which the corrected p-values were less than 0.05. Liver cirrhosis, the end-stage of every chronic liver disease including fibrosis, is a major risk factor for primary liver cancer [53]. Chronic inflammation status associated with liver cirrhosis can induce oxidative stress and alter the functions of the oxidant-generating enzymes and oncogenic proteins of the cells and thereby promote liver cancers formation [54]. Chronic inflammation can also facilitate angiogenesis and the growth, invasion, and metastasis of tumor cells to promote cancer development [55]. Fibrosis, the accumulation of collagens in the hepatic extracellular matrix (ECM), could retard the turnover of ECM and results in the activation of growth factor signaling cascades and cell proliferation in the liver, which promote cancer development. More than 80% of HCC, the most common type of liver cancer, develops in fibrotic or cirrhotic livers, suggesting the importance of the two conditions in promoting liver cancer development [56]. Recently, many bioinformatics approaches have identified several key genes and pathways for transforming cirrhosis to HCC, and Akr7a3, Cdc2a, and Cdkn3 were identified by these studies [42,50,51].

There are some limitations to the study. First, despite the differences in the definitions of the high doses, the data were grouped together in this study. The 5-days and 7-days exposure styles were also grouped as 1-week. Furthermore, the modulation of the genes may be affected by different study designs. Especially, dose level is critical when evaluating chemicals, and a high-dose level may increase specificity compared with a lower-dose level. A previous study also concluded that the optimal exposure style for assessing NGHCs is a 3-day daily high dose [57]. However, given that we are looking for time-invariant biomarkers and the prediction model performed well in the external database, the effect of the differences in the experimental study design should not be an issue. Secondly, animal studies are still needed for the application of the biomarkers for NGHC assessment. The biomarkers were derived from animal experiments. Despite largely shortening the length of the study, animal studies are still needed for the NGHC assessment. As for the concern of species differences between mice and rats, published studies have shown that the biomarkers derived from mice were applicable to rats for classifying genotoxic hepatocarcinogens, NGHCs, and NHCs [58,59]. Future works may be the investigation of species differences in the time-invariant biomarkers identified from this study. Thirdly, caution should be taken when applying the biomarkers or the model for interpretation of human data. Rodents and humans have inherent species differences so that the mechanisms of actions identified from NGHC exposure may not be applied in humans [60].

## 4. Conclusions

In summary, we have identified nine time-invariant biomarkers based on time-course gene expression data and further developed robust prediction models for NGHCs based on the time-invariant biomarkers. The analysis of the nine genes, namely A2m, Akr7a3, Aqp7, Ca3, Cdc2a, Cdkn3, Cyp2c11, Ntf3, and Sds, revealed the association between NGHCs and chronic inflammatory liver conditions, including liver cirrhosis and fibrosis. The time-invariant biomarkers derived from the short-term exposure styles were found to be more stable than the other biomarkers. The time-invariant biomarkers and the developed models could be reliable screening methods to prioritize chemicals of potential non-genotoxic hepatocarcinogenesis prior to the traditional 2-year rodent bioassays. The time-invariant biomarkers and their linkage to chronic inflammatory diseases provide a better understanding of the mechanisms of action for chemical-induced carcinogenicity in rodents and their relevance in human risk. From a public health standpoint, the time-invariant biomarkers are expected to improve the accuracy of the NGHC predictions from short-term animal studies, shorten the time and expense associated with the evaluation, and thereby accelerate the safety assessment for potential environmental pollutants and drug candidates. Metabolomics [61,62] may also be potential methods for identifying biomarkers for NGHCs. The integration of biomarkers from genes and metabolites might further improve the accuracy for NGHC identification.

## Figures and Tables

**Figure 1 ijerph-17-04298-f001:**
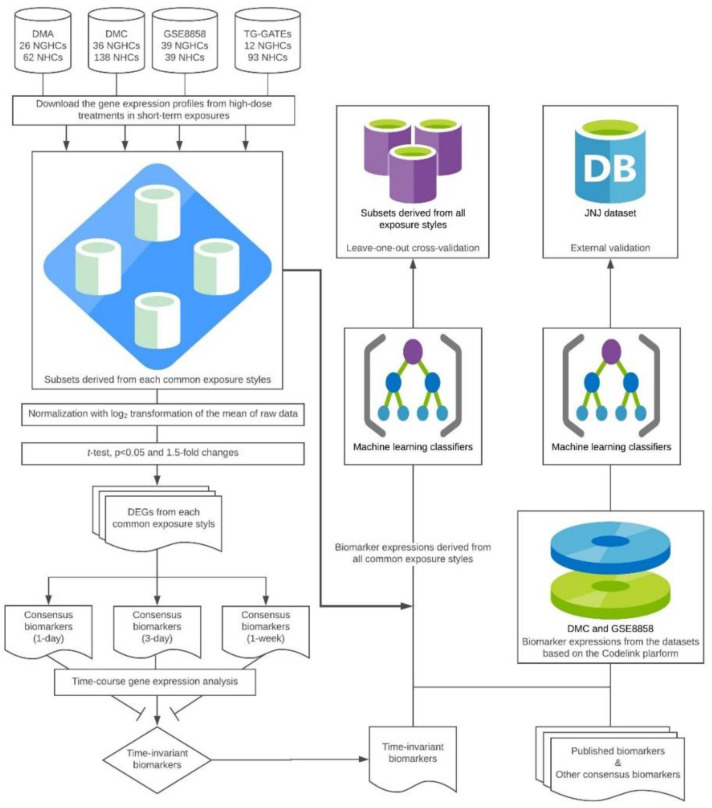
System flow. Abbreviations: DMA, Drug Matrix with Affymetrix platform; DMC, DrugMatrix with Codelink platform; DEGs, differential expression genes. * Consensus biomarkers (3 days) were identified as the time-invariant biomarkers; GSE8858, Gene Expression Omnibus accession no. 8858; TG-GATEs, Toxicogenomics Project-Genomics Assisted Toxicity Evaluation System; JNJ dataset, Johnson and Johnson dataset; NGHC, non-genotoxic hepatocarcinogens; NHC, non-hepatocarcinogens.

**Table 1 ijerph-17-04298-t001:** Biomarkers and corresponding modulations for three short-term exposure styles.

Biomarker Set	Gene Symbol	Affymetrix ID	Codelink ID	Modulation			Time-Invariant	Reference
				1D	3D	1W		
Consensus Biomarkers (1-day)	A2m	1367794_at	NM_012488	−	−	−	Yes	Huang and Tung (2017) [16]
	Ca3	1386977_at	NM_019292	−	−	−	Yes	
	Cxcl1	1387316_at	NM_030845	−	−	−	Yes	
	Cyp8b1	1368435_at	NM_031241	−	−	+/−	No	
Consensus Biomarkers (3-day) *	A2m	1367794_at	NM_012488	−	−	−	Yes	This study
	Akr7a3	1368121_at	NM_013215	+	+	+	Yes	
	Aqp7	1368317_at	NM_019157	+	+	+	Yes	
	Ca3	1386977_at	NM_019292	−	−	−	Yes	
	Cdc2a	1367776_at	NM_019296	+	+	+	Yes	
	Cdkn3	1372685_at	BE113362	+	+	+	Yes	
	Cyp2c11	1387328_at	NM_019184	−	−	−	Yes	
	Ntf3	1387267_at	NM_031073	−	−	−	Yes	
	Sds	1369864_a_at	NM_053962	−	−	−	Yes	
Consensus Biomarkers (1-week)	Akr7a3	1368121_at	NM_013215	+	+	+	Yes	This study
	Aqp7	1368317_at	NM_019157	+	+	+	Yes	
	Atf3	1369268_at	NM_012912	+	+/−	+	No	
	beta-sarcoglycan	1374796_at	AI413058	+	+	+	Yes	
	Ca3	1386977_at	NM_019292	−	−	−	Yes	
	Cpt1b	1367742_at	NM_013200	+	+	+	Yes	
	Cyp2c11	1387328_at	NM_019184	−	−	−	Yes	
	Cyp17a1	1387123_at	NM_012753	−	+/−	+	No	
	Ntf3	1387267_at	NM_031073	−	−	−	Yes	
	RGD1562428_predicted	1376296_at	BF387347	+	+	+	Yes	
	Snx10	1383585_at	AI043753	+	+	+	Yes	
E5	Abcb4	1369161_at	NA	−	−	−	Yes	Eichner et al. (2014) [13]
	Akr7a3	1368121_at	NM_013215	+	+	+	Yes	
	Ccng1	1367764_at	NM_012923	+	+	+/−	No	
	Cdkn1a	1387391_at	NM_080782	+	+/−	+/−	No	
	Phlda3	1375224_at	AW520812	+/−	+/−	+	No	
F19	Akr7a3	1368121_at	NM_013215	+	+	+	Yes	Fielden et al.(2011) [10]
	Aldh1a1	1387022_at	CK222590	+	+	+	Yes	
	Anxa2	1367584_at	AA956299	+/−	+	+	No	
	Btg2	1386994_at	NM_017259	−	+/−	+/−	No	
	Cdkn1a	1387391_at	NM_080782	+	+/−	+/−	No	
	Cited4	1390008_-at	NM_053699	+/−	+/−	+/−	No	
	ESTs	NA	BM388029	−	+	+	No	
	Gpr146	1373158_at	NA	−	−	−	Yes	
	Ica1	1367787_at	NM_030844	+	+	+	Yes	
	LitaF	1370928_at	U53184	+/−	+/−	+/−	No	
	Mat1a	1371031_at	X60822	−	−	−	Yes	
	Mgmt	1368311_at	NM_012861	+/−	+	+	No	
	Mt1a	1371237_at	CR458797	−	−	−	Yes	
	Ppia	1398850_at	BI303474	+/−	+/−	+/−	No	
	Prodh2	1389645_at	AI058310	−	−	−	Yes	
	Psmb9	1370186_at	NM_012708	+/−	+/−	+/−	No	
	Tap1	1388149_at	X57523	+/-	+/-	+/-	No	
	Trnt1	1383144_at	AI412002	+/-	+/-	+	No	
	Usp2	1387703_at	NM_053774	+/−	−	+/−	No	
U9	Abcb1a	1370583_s_at	NA	+	+	+	Yes	Uehara et al. (2011) [12]
	Acot9	1379262_at	NA	+	+	+	Yes	
	Cd276_1	1395737_at	BF398424	+/−	+/−	+	No	
	Cd276_2	1374198_at	NA	+/−	+	+	No	
	Cdh13_1	1375719_s_at	NM_138889	+/−	+/−	+/−	No	
	Cdh13_2	1373102_at	NA	+/−	+/−	+/−	No	
	Ica1	1367787_at	NM_030844	+	+	+	Yes	
	Tes	1383401_at	NM_173132	+	+/−	+/−	No	
	Tmem184c	1379419_at	NA	+	+	+	Yes	

ID: identity; 1D: 1-day; 3D: 3-day; 1W: 1-week; +: Upregulation of the gene by the non-genotoxic hepatocarcinogens (NGHCs) compared with the non-hepatocarcinogens (NHCs); −: Downregulation of the gene by NGHCs compared with NHCs; +/−: Inconsistent modulations of the biomarkers in the referenced datasets, NA: Not available, * Time-invariant biomarkers.

**Table 2 ijerph-17-04298-t002:** Performance of time-invariant biomarkers using different machine learning algorithms.

Algorithm	Performance (Median AUC from LOOCV)	Variance		
		IQR	C.V.d	C.V.e
Bagging Tree (BaT)	0.809	0.035	5.33%	6.17%
Boosting Tree (BoT)	0.757	0.102	9.37%	9.21%
Decision Tree (J48)	0.598	0.197	22.98%	24.35%
k-Nearest Neighbor (kNN)	0.720	0.071	5.81%	7.09%
Naive Bayes (NB)	0.800	0.055	3.50%	4.00%
Random Forest (RF)	0.817	0.041	4.36%	4.74%
Support Vector Machine (SVM)	0.582	0.084	8.56%	3.07%

Abbreviations: LOOCV, leave-one-out cross-validation; IQR, interquartile range; C.V.d, coefficient of variation from datasets; C.V.e, coefficient of variation from exposures; AUC, area under the receiver operating characteristic curve.

**Table 3 ijerph-17-04298-t003:** Performance of the time-invariant, consensus, and published biomarkers using Random Forest.

Signature	Dataset (Exposure Style)	Performance (Median AUC from LOOCV)		Variance (All Exposure)		
		Short-Term ^3^	All Exposure ^4^	IQR	C.V.d	C.V.e
Consensus biomarkers (1-day)	Multiple datasets ^1^ (1 day)	0.739	0.733	0.049 *	4.89%	4.02%
Time-invariant biomarkers /Consensus biomarkers (3-day)	Multiple datasets ^1^ (3 days)	0.817	0.824	0.036 *	4.34%	4.72%
Consensus biomarker (1-week)	Multiple datasets ^1^ (5 or 7 days ^2^)	0.780	0.810	0.111	7.41%	7.25%
E5	TG-GATEs (14 days)	0.656	0.656	0.097	9.04%	9.47%
F19	DrugMatrix (5 days)	0.796	0.809	0.085	6.47%	3.88%
U9	TG-GATEs (28 days)	0.703	0.713	0.057	8.25%	7.09%

Note: ^1^ DrugMatrix, GSE8858, and TG-GATEs; ^2^ five days exposure in DrugMatrix and GSE8858, and seven days exposure in TG-GATEs; ^3^ 1-day, 3- day, and 1-week high-dose exposures; ^4^ common short-term merged 14 days and 28 days high-dose exposures in TG-GATEs.* Significant difference (*p* < 0.05). A model with IQR, C.V.d, and C.V.e values less than 0.05 is considered as with a stable performance.

**Table 4 ijerph-17-04298-t004:** Performance of the time-invariant, consensus, and published biomarkers during external validation.

Signature	Performance (AUC from the Training Datasets)	
	DMC	GSE8858
Consensus biomarkers (1-day)	0.753	0.852
Time-invariant biomarkers /Consensus biomarkers (3-day)	0.862	0.857
Consensus biomarker (1-week)	0.820	0.815
E5	0.632	0.562
F19	0.732	0.791
U9	0.338	0.465

Abbreviations: AUC, area under the receiver operating characteristic curve; DMC, DrugMatrix with Codelink platform; GSE8858, Gene Expression Omnibus accession no. 8858.

**Table 5 ijerph-17-04298-t005:** Enriched Gene Ontology (GO) and disease terms of the time-invariant biomarkers.

Disease ID.	Disease Name	Involved Genes	Corrected *p*-Value *
MESH:D008106	Liver Cirrhosis (Experimental)	A2m, Aqp7, Ca3, Cdc2a, Cdkn3, Sds	2.97 × 10^−7^
MESH:D008103	Liver Cirrhosis	A2m, Aqp7, Ca3, Cdc2a, Cdkn3, Sds	6.52 × 10^−7^
MESH:D005355	Fibrosis	A2m, Aqp7, Ca3, Cdc2a, Cdkn3, Sds	9.82 × 10^−7^
MESH:D008107	Liver Diseases	A2m, Akr7a3, Aqp7, Ca3, Cdc2aCdkn3, Sds	1.40 × 10^−6^
MESH:D004066	Digestive System Diseases	A2m, Akr7a3, Aqp7, Ca3, Cdc2aCdkn3, Sds	1.61 × 10^−5^
MESH:D006528	Carcinoma, Hepatocellular	A2m Cdc2a, Cdkn3	0.015
MESH:D008113	Liver Neoplasms	A2m Cdc2a, Cdkn3	0.037

*, The corrected significance of the enrichment was adjusted for multiple testing using the Bonferroni method.

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
