# Peer review of "Identification of Time-Invariant Biomarkers for Non-Genotoxic Hepatocarcinogen Assessment"

_ijerph, 2020, doi:10.3390/ijerph17124298_

Round 1

Reviewer 1 Report

Substantial English and Grammar editing required, however an informative and cohesive manuscript. The following should be addressed prior to publication:

“AUC values of greater than or equal to 0857” should be …”0.857”?

Description of FR model implemented. Rationale.

Is Fig 2 the best way to present this data?

Conclusions need to be expanded to discuss the potential of the work, impact and future perspectives.

Additional references pertinent to this manuscript requiring insertion:

Kubyshkin, V. and Budisa, N., 2017. Amide rotation trajectories probed by symmetry. Organic & biomolecular chemistry15(32), pp.6764-6772.

Owen, L., Laird, K. and Wilson, P.B., 2018. Structure-activity modelling of essential oils, their components, and key molecular parameters and descriptors. Molecular and cellular probes38, pp.25-30.

Uehara, T., Hirode, M., Ono, A., Kiyosawa, N., Omura, K., Shimizu, T., Mizukawa, Y., Miyagishima, T., Nagao, T. and Urushidani, T., 2008. A toxicogenomics approach for early assessment of potential non-genotoxic hepatocarcinogenicity of chemicals in rats. Toxicology250(1), pp.15-26.

Leenders, J., Grootveld, M., Percival, B., Gibson, M., Casanova, F. and Wilson, P.B., 2020. Benchtop Low-Frequency 60 MHz NMR Analysis of Urine: A Comparative Metabolomics Investigation. Metabolites10(4), p.155.

Nicolaidou, V. and Koufaris, C., 2020. Application of transcriptomic and microRNA profiling in the evaluation of potential liver carcinogens. Toxicology and Industrial Health, p.0748233720922710.

Percival, B.C., Gibson, M., Wilson, P.B., Platt, F.M. and Grootveld, M., 2020. Metabolomic Studies of Lipid Storage Disorders, with Special Reference to Niemann-Pick Type C Disease: A Critical Review with Future Perspectives. International Journal of Molecular Sciences21(7), p.2533

Author Response

Response to Reviewer 1 Comments

Substantial English and Grammar editing required, however an informative and cohesive manuscript.

Response: Thank for the comment. We have done English and grammar editing through all text of the manuscript.

Point 1: “AUC values of greater than or equal to 0857” should be …”0.857”?

Response 1: Thanks for the correction. We have corrected the number as 0.857 and corresponding sentence is shown in the following.

“The application of the models to the external validation datasets achieved high AUC values of greater than or equal to 0.857” (Page 1, paragraph 1)

Point 2:  Description of RF model implemented. Rationale.

Response 2: Thanks for suggestion. The description of RF model implementation was appended as follows.

“The implementation of the RF algorithm was based on WEKA v3.8. The number of features for constructing a fully-grown decision tree was set to the default value of the square root of the number of features, genes of each biomarker set. The optimal number of trees ranging from 10 to 100 was determined based on the maximum AUC performance from the leave-one-out cross-validation (LOOCV).” (Page 4, paragraph 6)

Point 3: Is Fig 2 the best way to present this data?

Response 3:  Thanks for the suggestion. We have removed Fig 2 and appended a new Table 4 for better representation of the data (Page 10) as shown in the following.

Table 4. Performance of time-invariant, consensus and published biomarkers in external validation.

Signature

Performance (AUC from training datasets)

DMC

GSE8858

Consensus biomarkers (1-day)

0.753

0.852

Time-invariant biomarkers

/Consensus biomarkers (3-day)

0.862

0.857

Consensus biomarker (1-week)

0.820

0.815

E5

0.632

0.562

F19

0.732

0.791

U9

0.338

0.465

Point 4: Conclusions need to be expanded to discuss the potential of the work, impact and future perspectives.

Response 4: Thanks for suggestion. Additional sentences have been appended as follows to improve the manuscript.

“The time-invariant biomarkers and their linkage to chronic inflammatory diseases provide a better understanding of the mechanisms of action for chemical-induced carcinogenicity in rodents and their relevance in human risk. From a public health standpoint, the time-invariant biomarkers are expected to improve the accuracy of NGHC predictions from short-term animal studies, shorten the time and expense associated with the evaluation, and thereby accelerate the safety assessment for potential environmental pollutants and drug candidates. Metabolomics [61,62] may also be potential methods for identifying biomarkers for NGHCs. The integration of biomarkers from genes and metabolites might further improve the accuracy for NGHC identification.” (Page 12, paragraph 2)

Point 5: Additional references pertinent to this manuscript requiring insertion:

Response 5: Thanks for the suggestion to improve our work. The mentioned references have been cited.

(Page 2, paragraph 1)

  1. Kubyshkin, V. and Budisa, N., 2017. Amide rotation trajectories probed by symmetry. Organic & biomolecular chemistry, 15(32), pp.6764-6772.
  2. Owen, L., Laird, K. and Wilson, P.B., 2018. Structure-activity modelling of essential oils, their components, and key molecular parameters and descriptors. Molecular and cellular probes, 38, pp.25-30.
  3. Uehara, T., Hirode, M., Ono, A., Kiyosawa, N., Omura, K., Shimizu, T., Mizukawa, Y., Miyagishima, T., Nagao, T. and Urushidani, T., 2008. A toxicogenomics approach for early assessment of potential non-genotoxic hepatocarcinogenicity of chemicals in rats. Toxicology, 250(1), pp.15-26.
  4. Nicolaidou, V. and Koufaris, C., 2020. Application of transcriptomic and microRNA profiling in the evaluation of potential liver carcinogens. Toxicology and Industrial Health, p.0748233720922710.

(Page 12, paragraph 2)

  1. Leenders, J., Grootveld, M., Percival, B., Gibson, M., Casanova, F. and Wilson, P.B., 2020. Benchtop Low-Frequency 60 MHz NMR Analysis of Urine: A Comparative Metabolomics Investigation. Metabolites, 10(4), p.155.
  2. Percival, B.C., Gibson, M., Wilson, P.B., Platt, F.M. and Grootveld, M., 2020. Metabolomic Studies of Lipid Storage Disorders, with Special Reference to Niemann-Pick Type C Disease: A Critical Review with Future Perspectives. International Journal of Molecular Sciences, 21(7), p.2533.

Reviewer 2 Report

This article studies about non-genotoxic hepatocarcinogens (NGHCs) and explores the biomarkers for NGHCs using microarray analysis and machine-learning techniques in short-term exposure of NGHC chemicals. Analyzing the gene expression profiling of the public data, the authors found that gene expression of A2m, Akr7a3, Aqp7, Ca3, Cdc2a, Cdkn3, Cyp2c11, Ntf3 and Sds are not altered in time-dependent manner within short-term exposure less than 1 week administration. This study is interesting in terms of machine-learning based validation process evaluating the possibility of these biomarkers. The meaning of “time-invariant” may be explained more in detail, since this study only focuses on the short-term exposure of the NGHC chemicals. Table 1 needs to be revised to explain the modulation. It is not clear whether “-“ means low-expression of the genes or not. The rationale for calling low-expressed genes as biomarkers may be explained. The more detailed information on public data used in the article, such as the differences between mice and rats, or chemical exposure may be included. Figure 1 should be revised, since it seems that the texts are inverted. Significance in IQR in Table 3 may need more detailed explanation. Table 4 may need more detailed information such as data source of GO enrichment analysis.

Reviewer 3 Report

Non-genotoxic hepatocarcinogens (NGHCs) can only be confirmed by 2-year rodent studies. Toxicogenomics approaches using gene expression profiles from short-term animal studies could enable early assessment of NGHCs. In this study, 9 time-invariant biomarkers, namely A2m, Akr7a3, Aqp7, Ca3, Cdc2a, Cdkn3, Cyp2c11, Ntf3, and Sds were identified from four large-scale microarray datasets. Machine learning techniques were subsequently employed to assess the prediction performance of the biomarkers. The application of the models to the external validation datasets achieved high AUC values of greater than or equal to 0857. An enrichment analysis of the biomarkers inferred the involvement of chronic inflammatory diseases such as liver cirrhosis, fibrosis, and hepatocellular carcinoma to NGHCs. The authors concluded the time-invariant biomarkers provided a robust alternative for NGHC prediction.

  This is an interesting study. However, could the authors provide some examples and/or explanations about the applications of these time-invariant biomarkers in clinical practice or public health settings?

Author Response

Response to Reviewer 3 Comments

Non-genotoxic hepatocarcinogens (NGHCs) can only be confirmed by 2-year rodent studies. Toxicogenomics approaches using gene expression profiles from short-term animal studies could enable early assessment of NGHCs. In this study, 9 time-invariant biomarkers, namely A2m, Akr7a3, Aqp7, Ca3, Cdc2a, Cdkn3, Cyp2c11, Ntf3, and Sds were identified from four large-scale microarray datasets. Machine learning techniques were subsequently employed to assess the prediction performance of the biomarkers. The application of the models to the external validation datasets achieved high AUC values of greater than or equal to 0857. An enrichment analysis of the biomarkers inferred the involvement of chronic inflammatory diseases such as liver cirrhosis, fibrosis, and hepatocellular carcinoma to NGHCs. The authors concluded the time-invariant biomarkers provided a robust alternative for NGHC prediction.

Response: Thanks for the reviewer’s summary of our work.

Point 1: Could the authors provide some examples and/or explanations about the applications of these time-invariant biomarkers in clinical practice or public health settings?

Response 1: Thanks for suggestion to improve our work. To address the concerns, additional sentences have been appended as follows.

“Akr7a3 [51], Cdc2a [50,52], and Cdkn3 [50] have also been found as biomarkers for early diagnosis, staging, and prognosis in human liver cancer clinically.” (Page 11, paragraph 2)

“The time-invariant biomarkers and their linkage to chronic inflammatory diseases enhance our understanding to the mechanisms of action for chemical-induced carcinogenicity in rodents and their relevance in human risk. From a public health standpoint, the time-invariant biomarkers are expected to improve the accuracy of NGHC predictions from short-term animal studies, shorten the time and expense associated with the evaluation, and thereby accelerate the safety assessment for potential environmental pollutants and drug candidates.” (Page 12, paragraph 2)